# Non-Linear Frequency Dependence of Neurovascular Coupling in the Cerebellar Cortex Implies Vasodilation–Vasoconstriction Competition

**DOI:** 10.3390/cells11061047

**Published:** 2022-03-19

**Authors:** Giuseppe Gagliano, Anita Monteverdi, Stefano Casali, Umberto Laforenza, Claudia A. M. Gandini Wheeler-Kingshott, Egidio D’Angelo, Lisa Mapelli

**Affiliations:** 1Department of Brain and Behavioral Sciences, University of Pavia, 27100 Pavia, Italy; giuseppe.gagliano@unipv.it (G.G.); anita.monteverdi01@universitadipavia.it (A.M.); stefano.casali@unipv.it (S.C.); claudia.gandini@unipv.it (C.A.M.G.W.-K.); 2IRCCS Mondino Foundation, 27100 Pavia, Italy; 3Department of Molecular Medicine, University of Pavia, 27100 Pavia, Italy; umberto.laforenza@unipv.it; 4NMR Research Unit, Queen Square MS Centre, Department of Neuroinflammation, UCL Queen Square Institute of Neurology, Faculty of Brain Sciences, University College London, London WC1N3 BG, UK

**Keywords:** cerebellum, neurovascular coupling, granule cells, nitric oxide, NMDA receptor

## Abstract

Neurovascular coupling (NVC) is the process associating local cerebral blood flow (CBF) to neuronal activity (NA). Although NVC provides the basis for the blood oxygen level dependent (BOLD) effect used in functional MRI (fMRI), the relationship between NVC and NA is still unclear. Since recent studies reported cerebellar non-linearities in BOLD signals during motor tasks execution, we investigated the NVC/NA relationship using a range of input frequencies in acute mouse cerebellar slices of vermis and hemisphere. The capillary diameter increased in response to mossy fiber activation in the 6–300 Hz range, with a marked inflection around 50 Hz (vermis) and 100 Hz (hemisphere). The corresponding NA was recorded using high-density multi-electrode arrays and correlated to capillary dynamics through a computational model dissecting the main components of granular layer activity. Here, NVC is known to involve a balance between the NMDAR-NO pathway driving vasodilation and the mGluRs-20HETE pathway driving vasoconstriction. Simulations showed that the NMDAR-mediated component of NA was sufficient to explain the time course of the capillary dilation but not its non-linear frequency dependence, suggesting that the mGluRs-20HETE pathway plays a role at intermediate frequencies. These parallel control pathways imply a vasodilation–vasoconstriction competition hypothesis that could adapt local hemodynamics at the microscale bearing implications for fMRI signals interpretation.

## 1. Introduction

Neurovascular coupling (NVC) comprises mechanisms that link neuronal activity (NA) and vascular motility, adjusting the cerebral blood flow (CBF) to the energy demand of neural tissue. In the last couple of decades, increasing attention has been devoted to addressing NVC mechanisms, both in physiological and pathological conditions. The metabolic hypothesis, with the rapid glucose and oxygen consumption by active neural cells driving blood vessel responses, has long been considered the basis for the interpretation of blood oxygen level dependent (BOLD) signals recorded using functional magnetic resonance imaging (fMRI). However, the cellular and molecular mechanisms involved have not yet been clarified [1,2,3,4], and the validity of this hypothesis was undermined by several observations where CBF was not directly linked to changes in oxygen or glucose concentration [5,6,7]. Further studies supported the neurogenic hypothesis, in which CBF changes are determined by direct signaling through molecules released by neurons or glial cells [8,9,10]. Interestingly, the role of astrocytes in determining the vascular tone has been emphasized in the set-point hypothesis, balancing dilation and constriction to maximize vessel responses to NA [11]. Additional mechanisms as intrinsic vascular properties are involved as well [12] and might contribute to region-specific differences [13]. Finally, differences in vasodilation and vasoconstriction agents have been described throughout the brain [14]. It is highly likely that all these microscale mechanisms intervene to some extent in determining the CBF changes following neuronal activation, probably with different relevance depending on the region involved [7], complicating the interpretation of macroscale phenomena like the BOLD signal [3,15].

In the cerebellum, NVC shows particular properties. The densely packed granular layer is the main determinant of changes in cerebellar energy consumption and the granule cells, which are NOergic and release nitric oxide (NO) in both the granular and molecular layers [16,17], are likely to be the primary controllers of NVC [18,19,20]. Vasodilation in the granular layer is entirely mediated by neuronal N-methyl-d-aspartate receptor (NMDAR)-dependent production of NO, while the main vasoconstriction pathway is the metabotropic glutamate receptors (mGluRs)-dependent production of 20-hydroxyeicosatetraenoic acid (20-HETE), presumably released by astrocytes [21] and blocked by NO itself [8]. Vasodilation and vasoconstriction pathways likely converge on pericytes, in agreement with their role in controlling microvessel caliber and initiating the BOLD signal [22,23]. A possible NVC region specificity was recently suggested by the evidence of different BOLD responses in the cerebellar cortex during the execution of intensity-graded grip force motor task in humans, with the most substantial effects appearing in vermis lobule V and hemisphere lobule VI [24,25], but it is unclear whether these differences simply reflect different cortical inputs or local NVC factors. Moreover, reports of non-linear BOLD response alterations due to pathology such as multiple sclerosis make understanding NVC factors very important for explaining the mechanisms of disease [26,27].

Several studies reported a direct correlation between NA and blood perfusion, where NA is measured as the summed local field potentials (∑LFP) evoked during electrical or sensory stimulation [28,29,30,31,32,33,34,35,36]. Generally, the stimulus-evoked changes in CBF (or related parameters) linearly correlate with NA at increasing activation rates. Non-linearities were reported mainly as a ceiling-effect due to a saturation of CBF increase at certain frequencies or stimulus amplitudes [28,29,30,37,38], usually with increasing perfusion at increasing frequencies. Interestingly, few cases have been reported of perfusion decreasing at higher frequencies, well-explained by the corresponding decrease of ∑LFP [33,36]. In summary, the studies so far reported a linear correlation of blood perfusion with NA, with some non-linearity mainly due to structural limitations of vessel motility.

Here, we investigated NVC in the granular layer of acute cerebellar slices by electrically activating mossy fibers at 6 to 300 Hz and measuring capillary diameter changes and NA responses in the vermis lobule V and the hemisphere lobule VI. NVC increased with frequency but showed a marked inflexion at intermediate frequencies that was not attributable to saturation of the effect. An advanced realistic computational model of the granular layer was used to dissect the NA features best correlated with vessel responses, confirming the key role of NMDAR in determining the time course of vessel dilation. Unexpectedly, NA-related parameters (including ∑LFP) were not sufficient to explain the specific frequency-dependence profile of cerebellar NVC, suggesting that additional mechanisms might be involved. The hypothesis of an NA-driven frequency-dependent competition between vasodilation and vasoconstriction pathways was therefore proposed as a hypothesis and thoroughly discussed.

## 2. Materials and Methods

Animal maintenance and experimental procedures were performed according to the international guidelines of the European Union Directive 2010/63/EU on the ethical use of animals and were approved by the local ethical committee of the University of Pavia (Italy) and by the Italian Ministry of Health (authorization n. 645/2017-PR and following art.1, comma 4 of the D.Lgs.n. 26/2014 approved on 9 December 2017). According to this authorization, the sample size was estimated a priori using G*Power software (Wilcoxon–Mann–Whitney test, effect size 1.5, significance level 0.05) yielding an actual Power of 0.82. Reporting complies with the ARRIVE guidelines (Animal Research: Reporting in Vivo Experiments).

### 2.1. Preparation of Acute Cerebellar Slices

Acute 220 μm parasagittal cerebellar slices were obtained from juvenile (17–23 days old) C57BL/6 mice of both sexes, as reported previously [21,39,40,41] (see Appendix A for details), from both the vermis and hemisphere. Slices were recovered for at least 1 h in Krebs solution before being incubated for 1 h with 75 nM U46619 (Abcam, Cambridge, United Kingdom), a thromboxane agonist. For recording, slices were transferred to the recording chamber of an upright microscope (Slicescope, Scientifica Ltd., Uckfield, United Kingdom) or the HD-MEA (BioCAM X, 3Brain AG, Wädenswil, Switzerland), where Krebs solution was perfused (2 mL/min), maintained at 37 °C, and combined with 75 nM U46619. Krebs solution for slice cutting and recovery contained (in mM): 120 NaCl, 2 KCl, 1.2 MgSO_4_, 26 NaHCO_3_, 1.2 KH_2_PO_4_, 2 CaCl_2_, and 11 glucose, equilibrated with 95% O_2_-5% CO_2_ (pH 7.4). All drugs were obtained from Sigma-Aldrich (Merck KGaA, Darmstadt, Germany), unless otherwise specified.

### 2.2. Immunofluorescence Staining

Pericytes and capillaries were stained in the granular layers of the cerebellar vermis and hemisphere slices as previously described [21,42], focusing, respectively, on lobule V and lobule VI (see Appendix A for details). The rabbit anti-NG2 chondroitin sulfate proteoglycan (Millipore, Merck KGaA, Darmstadt, Germany) and FITC-isolectin B4 (Sigma-Aldrich, Merck KGaA, Darmstadt, Germany) primary antibodies were used to stain the pericytes and blood vessels, respectively. Slices were mounted on microscope slides, using ProLong^®^ Gold antifade reagent with DAPI (Molecular Probes, Thermo Fisher Scientifics, Waltham, MA, USA). Fluorescence of samples was observed with a TCS SP5 II LEICA confocal microscopy system (Leica Microsystems GmbH, Wetzlar, Germany) furnished with a LEICA DM IRBE inverted microscope. All acquisition files were visualized by LAS AF Lite software. Negative controls were carried out in parallel by treating slices with non-immune serum during the incubation procedures.

### 2.3. Time-Lapse Acquisition and Analysis of Capillary Diameter Changes

Granular layer vessels were identified using bright-field microscopy. Vermis lobule V and hemisphere lobule VI were first identified using a 4× objective (XL Fluor 4×/340, N.A.: 0.28, Olympus, Tokyo, Japan). Secondly, the granular layer was inspected using a 60× objective (LumPlanFl 60×/0.90 W, Olympus, Tokyo Japan) which allowed us to locate capillaries and pericytes. Only in-plane vessels (walls parallel to the acquisition system) with an inner diameter < 10 μm and surrounded by at least 1 pericyte (arrowheads in Figure 1B) were considered. In ex vivo conditions, capillaries lose intraluminal flow and pressure due to the mechanical stress of the slicing procedure. Slices were pre-incubated with 75 nM U46619, a thromboxane agonist that re-establishes the vascular diameter mimicking physiological conditions [8,11,42,43], as previously described [21]. Therefore, these measurements are the closest proxy for in vivo conditions. Once the focus of the objective was adjusted on the capillary walls, the pericyte soma was no longer clearly visible. With their long projections, pericytes regulate the caliber of distant portions of the vessels [23]. Only one capillary at a time was in focus using 60× magnification, allowing us to record only one capillary per slice, to avoid possible impairments in subsequent dilations given the absence of physiological conditions to maintain the tone [21]. There is no reason to believe that summing up observations from multiple slices might prevent the generalization of capillary dynamics in response to stimulation.

Mossy fibers were stimulated with 15 V stimuli (of 200 µs duration/pulse; corresponding to 50 µA given the electrode resistance) at 6, 20, 50, 100, and 300 Hz for 35 s, using a bipolar tungsten electrode (Warner Instruments, Holliston, MA, USA; electrode resistance 0.3 MΩ) in cerebellar lobule V or VI in vermis and hemisphere slices, respectively. In all cases, the mossy fibers stimulation induced a significant vasodilation (Figure 1B). Capillary responses in the granular layer were detected about 200 μm distant from the stimulating electrode placed on the white matter containing the mossy fibers (average distance in vermis slices: 131.93 ± 5.60 μm; *n* = 50 and hemisphere slices: 135.45 ± 5.28 μm; *n* = 49; vermis vs. hemisphere *p* = 0.648). It must be noted that the white matter bundle also contains climbing fibers and Purkinje cells axons. Since only mossy fibers contribute to granular layer responses, from this point onward we will indicate the stimulation procedure as “mossy fibers stimulation” in the rest of the manuscript. The time-lapse bright-field images of each capillary’s caliber changes were obtained using a CCD camera (DMK41BU, Imaging Source, Bremen, Germany), controlled by the IC-capture 2.1 software (Imaging Source, Bremen, Germany) to acquire 1 image every second (before, during, and after mossy fibers stimulation) with 5 ms exposure time. Image sequences were analyzed offline using the ImageJ software, as previously described [21,22]. The high sensitivity of the ImageJ measuring tool and the 60× objective used for image acquisition allowed us to reliably detect changes in vessel diameter in the submicrometer range. The portion of the vessel showing the maximal effect was considered for the analysis. The slope of the dilation curves was obtained by calculating the mathematical slope of the line passing through the points of percentage vessel dilations at 1 s and 5 s (Figure 2).

### 2.4. Electrophysiological Recordings of Neuronal Activity in Cerebellar Slices

Neuronal activity was recorded as local field potential (LFP) (Figure 3) in the granular layer of vermis lobule V and hemisphere lobule VI during mossy fibers stimulation. Slices were treated the same as for the time-lapse imaging and placed on the recording chamber of a high-density multi-electrode array (HD-MEA) system (BioCAM X, 3Brain AG, Wädenswil, Switzerland). The recording probe was a CMOS Arena biochip with 4096 microelectrodes arranged in a 64 × 64 matrix, covering an area of 2.67 mm × 2.67 mm. The electrodes’ size was 21 μm × 21 μm with a pitch of 42 μm. Neuronal activity was sampled at 17,840.7 Hz/electrode and acquired with BrainWave X (3Brain AG). Stimulation was provided by a tungsten bipolar electrode (Warner Instruments) placed over the mossy fibers using a micromanipulator (Patch-Star, Scientifica Ltd., Uckfield, United Kingdom). The stimulator unit was embedded in the BioCAM X hardware and set to deliver current pulses of 50 µA (200 µs duration/pulse), at 6, 20, 50, and 100 Hz, thus matching the same stimulation conditions used for the recordings of capillary responses. Due to technical limitations, the 300 Hz stimulation frequency was not included in the analysis. In the granular layer, neuronal response to mossy fibers stimulation originated LFPs. In both vermis and hemisphere, the LFPs showed a typical N_1_–N_2a_-N_2b_–P_2_ complex (inset in Figure 3): N_1_ corresponds to pre-synaptic volley activation, N_2a_-N_2b_ are informative of granular cells synaptic activation, and P_2_ is likely to represent currents returning from the molecular layer [44]. In order to characterize granule cell responses to stimulation, the analysis was focused on N_2a_ and N_2b_ peaks’ amplitude and time to peak. Recordings were exported and analyzed with MATLAB (MathWorks, Natick, MA, USA). Given the biochip electrodes’ properties and the densely packed neurons in the granular layer, a reasonable estimate of the number of neurons contributing to the signal was 10–15 cells per electrode (though it might be slightly underestimated, see Appendix A for details). To match the distances used for capillary responses, the signals recorded in the channels located beyond 200 µm from the stimulation electrode were not included in the analysis. Given the high spatial resolution of the HD-MEA used, the granular layer responses in a 200 µm range were detected by nine channels per slice. The average distance from the stimulating electrode of the electrophysiological responses was 133 ± 3 μm (*n* = 20 slices; *n* = 159 electrodes), not statistically different from the capillary average distances from the stimulating electrode in time-lapse imaging (133 ± 4 μm, *n* = 99 vessels; unpaired Student’s *t* test *p* = 0.99). The cumulative LFP during the stimulation was calculated to compare the NA trend to the capillary responses in the same conditions.

### 2.5. Computational Model of the Granular Layer

Simulations were performed using a detailed computational model which reproduced the anatomo-functional organization of the cerebellar granular layer (GL) network [45,46] endowed with biophysically realistic single cell models of granule cells (GrCs) and Golgi cells (GoCs), as well as a simplified reconstruction of mossy fiber terminals: the glomeruli (gloms). Detailed reconstruction of synaptic dynamics and receptors was also included, allowing us to reproduce the NMDA receptor-mediated current. The full GL network had a volume of 800 × 800 × 150 µm^3^; circuit organization and connectivity were reproduced according to specific connectivity rules, accounting for both geometric and statistical data (see also [47] for a similar approach). The vermis network included 384,000 GrCs, 914 GoCs and 29,415 gloms. For the cerebellar hemisphere, neuronal density distributions were adjusted according to cell data from the Allen Brain Atlas (https://mouse.brain-map.org/ (accessed on 12 November 2020)), with GrC and GoC densities in the hemisphere, respectively, 64% and 89% higher than in the vermis (while the GrCs/gloms ratio was assumed to be constant). This led to a network reconstruction of the hemisphere GL with 630,419 GrCs, 1733 GoCs and 48,291 gloms. It should also be noted that the GrCs/GoCs ratio changed significantly from the vermis (~420:1) to the hemisphere (~364:1).

The experimental protocol was reproduced stimulating a subset of gloms (in a sphere of 27.7 µm radius [45]) with 35 s trains at the different frequencies of 6, 20, 50, 100, and 300 Hz. The total amount of excited GrCs was 1290 and 1996 for vermis and hemisphere networks, respectively. Notice that the number of GrCs contributing to the simulated response is comparable to the estimated total number of GrCs contributing to the LFP signal (see above: 10–15 neurons per electrode, 9 electrodes per slice, 10 slices per condition, resulting in a number of neurons contributing to the total average LFP above 900–1350 units). NMDAR-mediated currents were recorded for each stimulated GrC. The cumulative sum of the NMDA-current, averaged over each cell, was calculated at 2, 20, and 35 s during stimulation.

The full description of the computational model is reported in Appendix A.

### 2.6. Model Validation

The model validation against experimental data was obtained using the N_2a_ peak of the granular layer’s LFP. While the N_2a_ peak is mainly informative of the AMPAR-mediated component of the granule cells’ response, the N_2b_ peak depends on both NMDAR’s activation and inhibitory inputs. Therefore, model validation is more solid against the N_2a_ peak, rather than a combination of different factors varying in an unpredictable manner. The model was validated using the following procedure. The average membrane depolarization of all the active GrCs was computed in the 1 ms time-window following the stimuli. This corresponded to the N_2a_ peak observed in the experiments. The procedure was repeated for all stimulation frequencies used in the experiments. The resulting set of measurements was normalized and low-pass filtered (Butterworth, 0.2 Hz, 2nd order). These simulated data were compared to experimental data obtained measuring the N_2a_ peaks.

### 2.7. Data Analysis

Data were compared using statistical paired and unpaired Student’s *t* test and with ANOVA test. First, we applied the Shapiro–Wilk test to check the normal distribution of data. Second, the parametric one-way ANOVA and, finally, the Fisher post hoc tests were used to validate the statistical significance. Data were considered statistically significant with *p* < 0.05 and were reported as mean ± SEM (standard error of the mean).

## 3. Results

In the cerebellum, CBF changes mostly rely on capillary diameter changes in the granular layer following changes in NA [21,22]. Synaptic activation determines the release of vasoactive molecules (NO and 20-HETE, [21]) acting on pericytes, contractile cells that enwrap the capillary wall [48,49]. Here, we analyzed the effect of different input frequencies on NVC in the granular layer of vermis lobule V and hemisphere lobule VI of the cerebellar cortex.

### 3.1. Anatomical Organization of Neurovascular Components in the Granular Layer of Vermis and Hemisphere

In the granular layer of both vermis and hemisphere, the microvessel walls consisted of tightly connected endothelial cells surrounded by granule cells (asterisks in Figure 1) but not by smooth muscle cells, in agreement with previous reports on rats [21]. Notably, granule cells are in close proximity with pericytes, making the latter very suitable to receive chemical transmitters from activated neurons [50]. In both vermis and hemisphere, cerebellar slices showed granular layer capillaries labeled by anti-isolectin B4 and anti-proteoglycan NG2 primary antibodies, which stained vessel and pericyte membranes, respectively [21,42] (Figure 1A). The pericytes appeared to enwrap capillaries (arrowheads in Figure 1A,B), whose pre-contracted average internal diameter was not statistically different (unpaired Student’s *t* test, *p* = 0.13) in the vermis (2.75 ± 0.15 µm, *n* = 50) and hemisphere (2.47 ± 0.11 µm, *n* = 49). Thus, the cerebellar vermis and hemisphere showed similar granular layer organization of neurovascular components, composed of capillaries, pericytes, and granule cells.

### 3.2. Non-Linear Frequency-Dependent Dilation of Granular Layer Capillaries (NVC)

NVC was assessed by measuring capillary diameter changes following mossy fibers stimulation. Mossy fibers stimulation determined a significant vasodilation in granular layer capillaries, in both the vermis and hemisphere, at all the frequencies tested (6, 20, 50, 100, and 300 Hz; Appendix A) but the amount of dilation varied (Figure 1C and Figure 2). In the vermis (Figure 1C and Figure 2A), significant differences were evident at 2 s (20 Hz vs. 50 Hz), 20 s (20 Hz vs. 50 Hz) and 35 s (50 Hz vs. 100 Hz; 6 Hz vs. 20 Hz, 100 Hz, 300 Hz). In the hemisphere (Figure 1C and Figure 2A), significant differences were evident at 2 s (50 Hz vs. 300 Hz) and 35 s (100 Hz vs. 20 Hz, 300 Hz). Consequently, the dilation did not increase linearly with frequency, as illustrated by the points taken at 35 s (Figure 2B).

The time at which the maximum dilation was reached varied for each vessel, with an average of 23.7 ± 1.5 s in the vermis and 21.4 ± 1.4 s in the hemisphere (*n* = 50 and 49, respectively, *p* = 0.274). To exclude the effect of time, we considered the maximum dilation at the different frequencies, dissociating the effect of frequency from the effect of time. This comparison did not reveal any statistical difference between the two methods (*p* = 0.73 and *p* = 0.76 for the vermis and hemisphere, respectively) and confirmed the presence of a significant inflection in the dilations observed at 50 Hz and 100 Hz for the vermis and hemisphere, respectively (*p* = 0.022). E.g., at 100 Hz, dilation in the vermis was at its maximum (at 35 s 7.25 ± 1.27%, *n* = 10), while in the hemisphere it was around minimum values (at 35 s 3.21 ± 1.29%, *n* = 10, *p* = 0.032). It should be noted that the maximum dilation was similar in the vermis and hemisphere, despite differences in the frequency-dependence profile. To provide the basis for a comparison to the BOLD signals (peaking in few seconds) we calculated the slope of the dilation curves in the first 1–5 s during stimulation. Interestingly, the 1–5 s slope correlated well with maximum dilation, both in the vermis and hemisphere (Figure 2D). The plot of the slope as a function of stimulation frequency also showed the same non-linear trend observed when measuring dilation at 35 s and maximum dilation (Figure 2E).

### 3.3. Granular Layer Responses to Mossy Fibers Stimulation (NA)

In order to correlate NA with blood vessel dilation, several studies in different brain areas used local field potential (LFP) cumulative amplitude [28,29,30,31,32,33,34,35,36]. Here, the LFPs elicited in the granular layer by mossy fibers stimulation were measured using a high-density multi-electrode array (HD-MEA). This technique allowed us to assess the spatial distribution of NA and to record the LFPs at a distance from the stimulating electrode comparable to that of the capillaries. In this way, NA was recorded by the same tissue volume that was likely to generate the bulk of the signals that correlate NA to NVC (see Methods; Figure 3A).

According to previous characterizations of granular layer LFPs [44,51], N_2a_ and N_2b_ peaks derive from the synaptic activation of multiple granule cells clustered around the electrode. N_2a_ mainly depends on the AMPAR-mediated component of granule cell responses and on spike synchronicity, while N_2b_ is informative on the NMDAR-mediated component of granule cell responses and its inhibitory control [44,52].

The response to single-pulse mossy fibers stimulation was first characterized in the granular layers of the vermis and hemisphere (Figure 3A). In the vermis, N_2a_ peaked at 1.59 ± 0.03 ms with an amplitude of −158.74 ± 5.53 µV and N_2b_ peaked at 4.36 ± 0.10 ms with an amplitude of −67.93 ± 3.04 µV (*n* = 10 slices, 81 electrodes for all measurements). In the hemisphere, N_2a_ peaked at 2.05 ± 0.06 ms with an amplitude of −137.39 ± 4.66 µV and N_2b_ peaked at 3.79 ± 0.09 ms with an amplitude of −78.88 ± 2.90 µV (*n* = 10 slices, 76 electrodes for all measures). Thus, N_2a_ had a higher peak and shorter time-to-peak in the vermis compared to the hemisphere (unpaired Student’s *t* tests, *p* = 0.003 and *p* = 6.85 × 10^−10^, respectively); conversely, N_2b_ had a higher peak and shorter time-to-peak in the hemisphere compared to the vermis (unpaired Student’s *t* tests, *p* = 0.01 and *p* = 0.000123, respectively) (Figure 3B).

In the same slices, granular layer responses were characterized using the same stimulation patterns previously used to assess vasodilation (except for 300 Hz, see Methods). The time course of N_2a_ peak amplitudes at different frequencies for vermis and hemisphere is reported in Figure 4A. In all cases, during the stimulation, N_2a_ peak amplitudes decreased and attained a plateau. The decrease was more evident at higher frequencies (see Appendix A for the percent change at the end of the stimulation). N_2b_ peak amplitudes showed a trend similar to N_2a_ (Figure 4A). It should be noted that, despite identical stimulation intensity, N_2a_ was usually smaller in the hemisphere than in the vermis (e.g., at 2 s, 20 s, 35 s; *n* = 10 for both, unpaired Student’s *t* test *p* < 0.05).

The LFP amplitude was considered as a proxy of NA. However, N_2a_ and N_2b_ showed an almost exponential decrease during the stimulus trains, mostly reflecting adaptation in the neurotransmission process [53], with time-constants that became smaller at higher frequencies. This trend markedly differed from that of NVC (cf. Figure 1C), so that neither N_2a_ nor N_2b_ nor their sum explained the time course of dilation and its non-linearity in the frequency domain (Figure 4B).

### 3.4. Simulated NMDA Currents Correlate with NVC Time-Course

The biochemical pathway leading to capillary dilation in the granular layer relies on the activation of postsynaptic NMDARs in granule cells causing neuronal nitric oxide synthase (nNOS) activation and NO release [21]. Since the NMDA LFP component could not be quantitatively extracted from N_2b_ (not being the only contributor to this peak), we used a realistic computational model of the cerebellar network [45] to simulate the NMDAR component of granular layer responses to mossy fiber inputs. The model of the vermis was the same used previously [45,47], while the model of the hemisphere was adjusted to tune cell densities according to the Allen Brain Atlas (see Methods). Accordingly, granule cell density was increased resulting in a higher granule cell/Golgi cell ratio (Figure 5A). Model validation was performed against the experimental data at different time points and frequencies, showing a high level of convergence (Figure 5B; see also Methods for details). Eventually, the model yielded the NMDA current build-up in granule cells during the different stimulation patterns in the vermis and hemisphere.

We then considered that vasodilation in the granular layer is mediated by NMDAR-dependent production of NO [21] acting, through volume diffusion, on guanylyl cyclase (GC) in pericytes. The consequent cyclic guanosine monophosphate (cGMP) levels, which are the ultimate cause of vasodilation, correlate almost linearly with NO levels [54]. However, cGMP levels decrease due to the action of phosphodiesterase (PDE) that activates with a slow time-constant (around 20 s) [54]. This eventually decreases cGMP concentration by 26.4% and 59.9% after 20 s and 35 s, respectively [54]. Estimating the impact of NMDAR activation on NVC then requires a correction for PDE action. We, therefore, simulated the cumulative NMDA current build-up, corrected by PDE (−26.4% and −59.9% after 20 and 35 s, respectively [54]), and used it as a proxy of the signal controlling NVC (Figure 6A). Indeed, the corrected NMDA current build-up approached NVC, at all frequencies tested (Figure 6B and Appendix A).

In aggregate, these simulations suggested that the NMDAR-NO-GC pathway, incorporating the effect of PDE on cGMP levels, is sufficient to explain the time course of vasodilation, at each frequency tested, for both vermis and hemisphere. Conversely, NMDAR activity alone was not sufficient to explain either the frequency dependence or region dependence of NVC (Figure 6C to Figure 2B; the two trends reported, normalized for the amplitude scale, were statistically different with *p* = 0.013 and *p* = 0.00003 for the vermis and hemisphere, respectively).

## 4. Discussion

The central finding in this study was that NVC in the cerebellum shows non-linear frequency dependence. The capillary diameter increased in response to mossy fiber activation in the 6–300 Hz range, with a marked inflection around 50 Hz (vermis) and 100 Hz (hemisphere). Notice that the physiological average frequency of mossy fiber discharge is below 200 Hz [55], but they have been reported to sustain brief burst of activity with instantaneous frequencies up to 700 Hz [56,57,58]. For this reason, we also tested the ultrafast band, though its physiological relevance is less clear. In any case, the main finding of this study concerned the physiological 50–100 Hz frequency range. Interestingly, the NMDA receptor-mediated component of the granular layer responses triggering nNOS activation and NO production, once corrected for PDE activation (Figure 7A), could effectively explain the time course of vessel dilation at each stimulus frequency but was insufficient to explain the non-linearity in vessel response with respect to input frequency. These results revealed a complexity and diversity of NVC at the microscale suggesting a partial uncoupling from NA and the intervention of other mechanisms in addition to the NMDA-nNOS-NO pathway.

### 4.1. Non-Linearity and Region Specificity of Cerebellar NVC

Mossy fiber stimulations induced capillary vasodilation in the mouse cerebellar granular layer, attaining, on average, a maximum after 22 s. This effect, which was observed both in the vermis and the hemisphere, resembles that previously reported in rat cerebellar slices [21]. Interestingly, vasodilation reached the maximum at 20 Hz, but then decreased only to rise back to maximum at 300 Hz. This observation was confirmed using the dilation at the end of the stimulation (35 s), maximum dilation (independent from time), and the slope of the dilation curves in the first 1–5 s. These data demonstrate that cerebellar NVC is non-linear with respect to circuit stimulation frequency. The NVC curve, after bending beyond 20 Hz, recovered more rapidly in the vermis than in the hemisphere. It is worth noting that capillary dilation ranging from ~2.5% to ~7.5% is sufficient to increase the total blood flow by 4.8–16.5%, and the cerebrovascular volume by 2.4–7.9% [21,43,59] within the physiological range associated with BOLD responses in rodents in vivo [60,61] (see Appendix A for details). The fact that the initial slope of dilation was predictive of the maximum dilation in the different conditions offers a background to fill the gap between the time resolution of BOLD signals (peaking in a few seconds) and ex vivo investigations of NVC (characterized by longer time-constants to reach the maximum effect).

The reasons for the region specificity of cerebellar granular layer NVC remain unclear. In both the granular layer of cerebellar vermis and hemisphere, the pericytes were detected on capillary walls and in close contact with granule cells, i.e., in the ideal location to detect the chemical mediators released by neurons and glial cells in response to local NA changes [22,50]. Thus, the fundamental anatomical organization of the main cellular components was similar. Some differences in NA (relative size of the N_2a_ and N_2b_ LFP components) were observed between the two regions but did not explain those in NVC. Thus, neither anatomical nor electrophysiological data could explain local NVC differences. Therefore, there is a partial uncoupling of NA from NVC, and metabolic effects may play a role along with the neurogenic ones [7]. This case resembled that reported in the cerebral cortex, where differences in CBF occurred between the somatosensory and frontal areas of awake mice during locomotor activity on a treadmill [62] in the absence of differences in NA.

### 4.2. NA and NVC in the Cerebellar Vermis and Hemisphere

Reports in various brain areas have identified the ∑LFP as the measure of NA that best matches NVC [28,29,30,31,32,33,34,35,36]. Here, we used a high-density multi-electrodes array (HD-MEA) to record LFPs from the granular layer of cerebellar cortical slices in the vermis and hemisphere during mossy fibers stimulation. In both the vermis and hemisphere, the LFP was composed of the typical N_2a_-N_2b_ wave sequence [44,51,52], which is informative on the AMPAR- and NMDAR-mediated components of granule cell responses to mossy fibers stimulation, respectively. Interestingly, N_2a_ was larger in the vermis while N_2b_ was larger in the hemisphere. These differences were well-matched by updating the granule cell-Golgi cell ratio and neuronal density in the hemisphere and the vermis according to the Allen Brain Atlas in a realistic microcircuit model and could, therefore, reflect differences in local network organization. We expected that the differences in NA could explain the NVC non-linearity. However, the LFP parameters did not account for NVC dynamics, neither in the time nor frequency domains.

### 4.3. NMDAR Currents Drive the Time Course of Dilation but Do Not Determine Its Frequency Dependence

Unlike other brain regions, capillary vasodilation in the cerebellar granular layer is known to rely entirely on the NMDAR-nNOS-NO pathway, which is in line with the fact that this layer has the highest level of nNOS expression in the brain [14,21]. Moreover, a previous report in the rat somatosensory cortex showed the failure of ∑LFP in accounting for CBF variations at specific frequencies when the NMDA currents did not contribute to the signal [29]. It should not be surprising, then, that the parameter describing the relationship between NA and the time course of dilation in our case is the NMDAR-mediated component of granule cell response. In this study, a realistic theoretical model of the granular layer, tuned against subtle anatomical differences between vermis and hemisphere, was crucial to reconstruct the NMDAR-mediated component of the response. Following validation against experimental data, the simulated NMDA current build-up proved suitable to determine the time course of vasodilation. Nevertheless, the NMDA current could neither explain the frequency dependence of vasodilation (see Figure 2B) nor its regional specificity.

### 4.4. The Vasodilation–Vasoconstriction Competition Hypothesis

In a previous study [21], we demonstrated that NVC in the granular layer is mediated by NA-dependent release of the vasodilator agent NO, which occludes the effect of the vasoconstrictor agent 20-HETE, released following mGluRs activation, presumably in glial cells. The effect of 20-HETE appeared only when NOS activity was blocked, in line with the notion that NO blocks this vasoconstrictor pathway [8] (Figure 7A). The inflection in the NVC-frequency plot hints at a competition between vasodilator and vasoconstrictor pathways based on the frequency-dependent balance between NO and 20-HETE production (Figure 7B). Indeed, in rat cerebellar slices, pharmacological subtraction experiments revealed that 20-HETE reduced the vasodilation mediated by NO at 50 Hz by 44%, in fair agreement with the data reported here [21].

The vasodilation–vasoconstriction competition hypothesis is illustrated in Figure 7C. The amount of NO produced at low frequency (<50 Hz) is sufficient to block 20-HETE synthesis. Indeed, 20-HETE depends on the mGluRs activation, which needs neurotransmitter accumulation during high-frequency input discharges [63]. At increasing frequencies, glutamate build-up and spillover would intensively activate extrasynaptic mGluRs [63,64] recruiting the 20-HETE pathway. At this point, 20-HETE synthesis would surge, NO would not be able to counteract it, and the pericytes would combine the two signals causing the inflection of NVC curves observed at intermediate frequencies (50–150 Hz). At high frequency (>50–100 Hz), NO would increase enough to block the 20-HETE synthesis. Indeed, astrocytes mGluRs have been reported to mediate an intracellular Ca^2+^ signal which saturates, at increasing glutamate concentrations, besides showing desensitization [63,65,66]. This might help in explaining the decrease in the vasoconstriction efficiency at increasing stimulation frequencies. To be noted, the mGluRs are G-protein-coupled receptors and the pathway for 20-HETE synthesis involves more molecular steps [65,66,67,68] than that of NO production (NMDARs –> nNOS –> NO), which peaks in about 200 ms from stimulus onset [21]. Therefore, the release kinetics of 20-HETE may be somehow slower than those of NO, but this is probably irrelevant to the present case, given the 10 times slower timescale (seconds) of vessel dilation. This reasoning may as well apply to the fMRI recordings in vivo (with a sampling rate on the scale of seconds).

Although not exhaustive, this set of mechanisms accounts for the NVC/frequency non-linearity and provides the basis for a future quantitative mathematical model. Needless to say, such a model would provide a means to understanding neurological and neurodegenerative diseases, such as multiple sclerosis, where the non-linear BOLD response to a variable grip-force task was altered in the primary motor cortex [27]. Future work should aim not only to improve the mathematical model explaining the NVC complexity from cellular physiology experiments, but to scale such models to interpret in vivo data from human studies of central nervous system pathologies [69].

### 4.5. Comparing Different NVC Hypotheses and the Case for Cerebellar Region Specificity

The vasodilation–vasoconstriction balance, different from an NA ceiling effect, could prevent the system from saturating through a dynamic regulation of the vessel caliber set-point when the input frequency increases. However, it does not account alone for the shift from 50 to 100 Hz in the curve inflexion between the vermis and hemisphere and requires further comments. We could not rule out that subtle anatomical differences occur beyond the resolution of this study, addressing the “neurogenic hypothesis” of NVC. Other factors that are likely to come into play have been reported in different brain regions. Activity-dependent metabolites and the lowering of glucose and oxygen concentrations can link neuronal activity and vessel tone (e.g., see discussion in [7]) addressing the “metabolic hypothesis”. It is also interesting to consider the “set-point hypothesis” for determining dilation or constriction depending on the initial vascular tone [11]. Although we did not find any difference in pre-constricted capillary diameter between vermis and hemisphere, we could not rule out the possibility that different set-points characterize these two regions, so that similar initial diameter would not imply the same dilation/constriction balance. In addition, the cells combining multiple signals could include not just the pericytes but also the astrocytes [11] and endothelium [12], although blood vessel intrinsic properties should have been marginally involved due to the lack of intravascular pressure in acute brain slices. In summary, it is highly probable that NVC does not simply rely on the metabolic or neurogenic hypotheses but is defined by a complex interaction of multiple factors that vary locally from one brain region to another. 

### 4.6. Considerations on the Development of Cerebellar NVC

Possible changes in NVC during development were proposed as an explanation for the negative BOLD signals characterizing infants at early developmental stages [70,71]. Though little is known about the development of cerebellar NVC, we used juvenile mice in our study and a few considerations might be needed. First, granule cells and synaptic connections in the granular layer are functionally mature at this stage (in terms of ionic channels, synaptic transmission mechanisms, and plasticity) [41,72,73,74,75,76]. Second, the organization of granular layer astrocytes around the glomeruli is properly structured and functional in juvenile rodents [77,78]. It should be noted that cerebellar astrocytes are quite different from brain astrocytes [79], and their proliferation significantly drops between 4 and 7 days after birth [80]. Thirdly, we did not find any age dependence in capillary dilation in our data. Although there is no evidence that NVC might have expressed immature properties in our study, the understanding of the neurovascular unit in the cerebellum is far from complete and a possible influence of a late phase of development cannot be excluded.

## 5. Conclusions

The mechanisms of NVC and its impact on local brain functioning are gaining increasing attention in the scientific community, both for understanding BOLD signals used in fMRI and for the potential implications in neurovascular pathology. Our data showed that NVC is probably more complex than previously thought, adjusting to the input frequency in a region-specific manner. To the best of our knowledge, this was the first report of this effect in the cerebellar cortex. Interestingly, our results suggested that the force/BOLD non-linearity recorded from the cerebellum during motor task execution [24,25] could, at least in part, be due to local non-linear NVC properties. Although the understanding of NVC fine-tuning is still incomplete, our results integrated the neurogenic and metabolic hypothesis opening the perspective of a dynamic microvessel diameter regulation, which should be considered to interpret BOLD signals in fMRI recordings.

## Figures and Tables

**Figure 1 cells-11-01047-f001:**
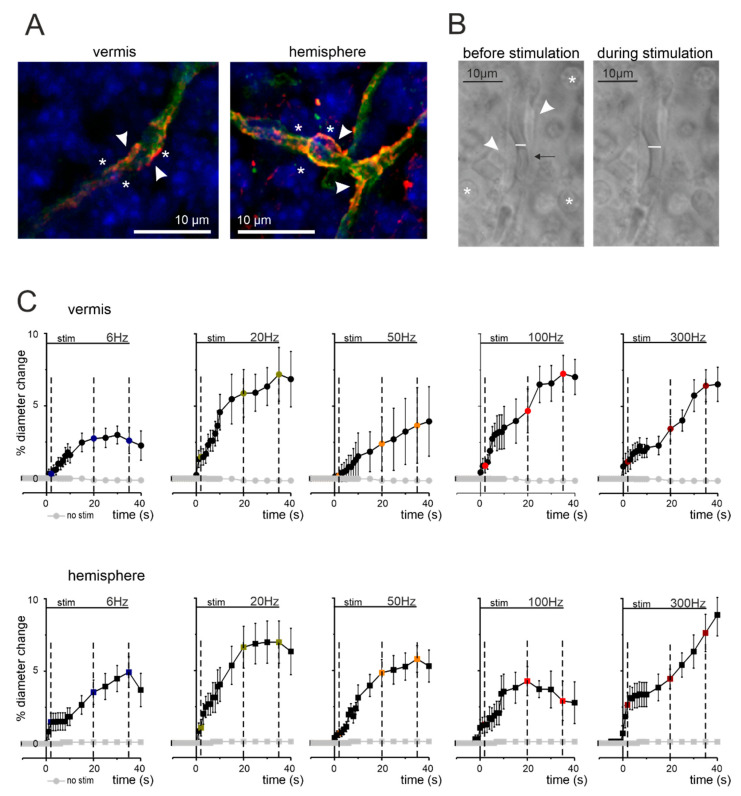
Anatomo-physiological correlations of NVC in the cerebellar granular layer. (**A**) Confocal fluorescent images of cerebellar slices stained for IB4 (green), NG2 (red), and DAPI (blue) to identify capillaries, pericytes, and cell nuclei, respectively. In both vermis and hemisphere, granule cells (asterisks) are in proximity of pericytes (arrowheads) located on capillary endothelium (green). (**B**) Bright-field images taken before and during mossy fiber stimulations at 50 Hz (vermis). The pericytes (arrowheads) are visible near granule cells (asterisks) and the capillary walls can be easily identified. The internal diameter changes in response to stimulation (white bar). The arrow indicates a red blood cell that moves inside the capillary as a consequence of blood vessel motility. (**C**) Average time courses of capillary dilation during mossy fibers activation at different frequencies in vermis and hemisphere. The figure shows the average percent change in capillary diameter size in the different conditions tested for the vermis lobule V and hemisphere lobule VI. Mossy fibers were stimulated for 35 s at 6, 20, 50, 100, and 300 Hz (stim bar). The dashed lines indicate the time points at 2, 20, and 35 s, used to analyze dilation in the subsequent analysis. In each panel, the dilation at the end of stimulus was significantly different compared to pre-stimulus baseline (see Appendix A). In both panel sets, grey dots represent stability recordings in which stimulation was not delivered (*n* = 8).

**Figure 2 cells-11-01047-f002:**
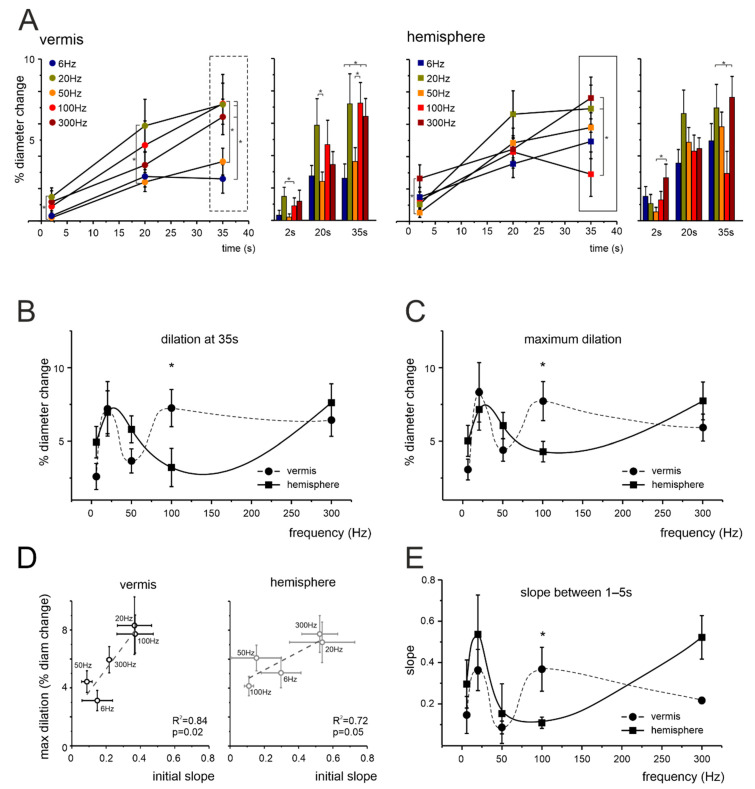
NVC frequency dependence in the cerebellar granular layer. (**A**) Average time course of vasodilation during mossy fibers stimulation at 2, 20, and 35 s in capillaries of the vermis and hemisphere. Asterisks indicate pairs of points that were statistically different (* *p* < 0.05). Vermis: 2 s, 20 Hz vs. 50 Hz (*p* = 0.039); 20 s, 20 Hz vs. 50 Hz (*p* = 0.0358); 35 s, 6 Hz vs. 100 Hz (*p* = 0.014), 50 Hz vs. 100 Hz (*p* = 0.049), 6 Hz vs. 20 Hz (*p* = 0.015), 6 Hz vs. 300 Hz (*p* = 0.036) (one-way ANOVA; for increasing frequencies *n* = 9, 10, 10, 10, 11, respectively). Hemisphere: 2 s, 50 Hz vs. 300 Hz (*p* = 0.029); 35 s, 100 Hz vs. 20 Hz (*p* = 0.024), 100 Hz vs. 300 Hz (*p* = 0.009) (one-way ANOVA; for increasing frequencies *n*= 9, 10, 10, 10, 10, respectively). The box markers show the data that are used for plots in (**B**), unfolded along the *x*-axis. (**B**) Average dilations at 35 s at each frequency tested in the vermis and hemisphere (dots and squares, respectively, connected by trend lines). The dilations in the vermis and hemisphere were statistically different at 100 Hz (*p* = 0.039). (**C**) Same plot as in (**B**), using the maximum dilation instead of the dilation at a fixed time point. The dilations in the vermis and hemisphere were statistically different at 100 Hz (*p* = 0.022). (**D**) The plots show the positive correlation between the maximum dilation at each frequency and the slope in the first 1–5 s, for the vermis (left) and hemisphere (right). (**E**) Same plot as in (**B**,**C**), using the slope in the first 1–5 s as a function of stimulation frequency. The slopes in the vermis and hemisphere were statistically significant at 100 Hz (*p* = 0.031).

**Figure 3 cells-11-01047-f003:**
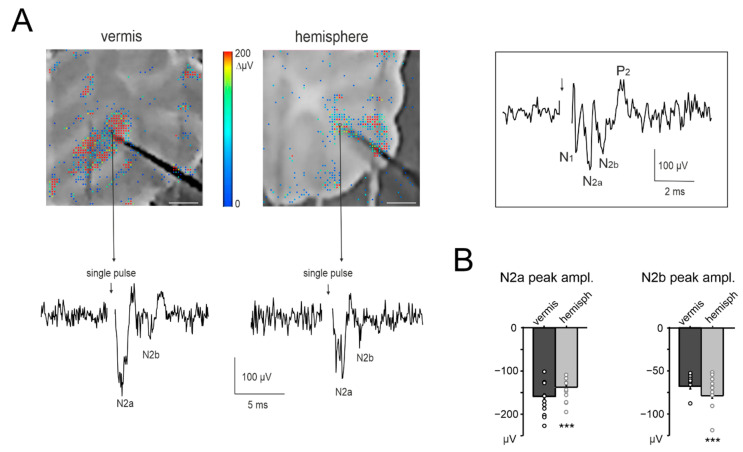
NA recordings in the cerebellar granular layer. (**A**) Example of a vermis and hemisphere cerebellar slice placed on the HD-MEA biochip (scale bar 0.5 mm). The activity following single-pulse stimulation of mossy fibers is superimposed in a color-scale (2 ms bin). A representative trace is shown from the HD-MEA biochip channel indicated by the arrow. The stimulus intensity and stimulating electrode is the same as used in Figure 1B. The inset shows the entire N_1_-N_2a_-N_2b_-P_2_ complex that comprises the classic LFP in the granular layer (see Methods for details). (**B**) The histograms show the average peak amplitude of N_2a_ and N_2b_, for both the vermis and hemisphere (*n* = 10 slices for both) and the single data points. Statistical significance is indicated by asterisks (*** *p* < 0.001).

**Figure 4 cells-11-01047-f004:**
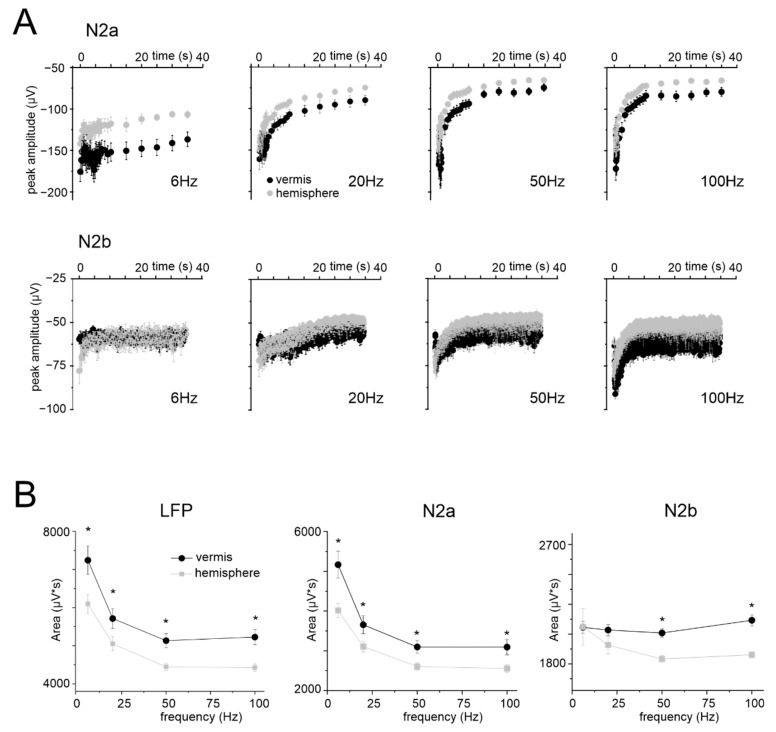
Time and frequency dependence of LFPs during stimulation. (**A**) The plots show the time course of N_2a_ and N_2b_ peak amplitude in response to stimulation at different frequencies, recorded in 10 vermis and 10 hemisphere slices (the responses in the right part of the N_2a_ plots are down-sampled). (**B**) Frequency dependence of the average cumulative LFP response, N_2a_ and N_2b_ measured at 35 s during stimulation at different frequencies in the vermis (*n* = 10) and hemisphere (*n* = 10) (same slices as in panel (**A**)). Statistical significance is indicated by asterisks (* *p* < 0.05).

**Figure 5 cells-11-01047-f005:**
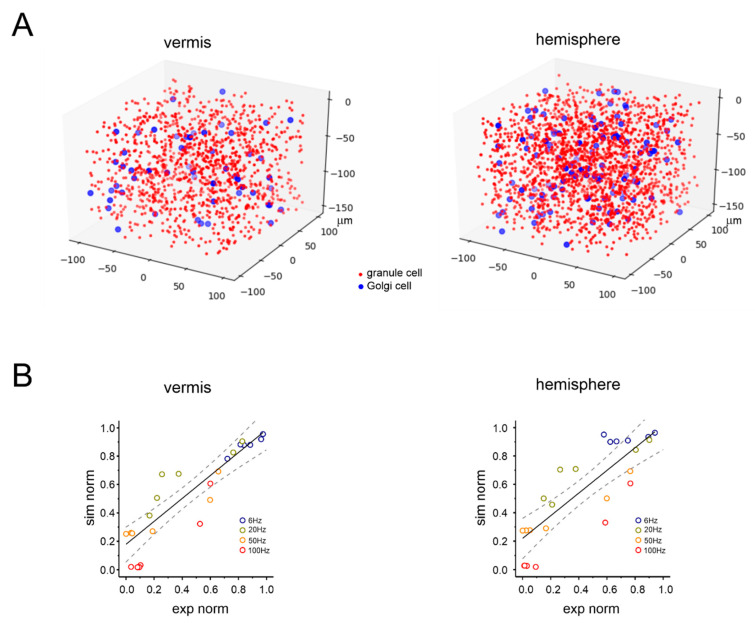
Realistic computational modeling of granular layer response in the vermis and hemisphere. (**A**) Schematic illustration of the spatial distribution of granule cells (red dots) and Golgi cells (blue dots) in a volume chunk of the granular layer (100 µm^3^) of the cerebellar vermis and hemisphere in a realistic computational model derived from 45. For better rendering, granule cells are down-sampled by a factor of 20. There are 1203 granule cells and 55 Golgi cells for the vermis, 1980 granule cells and 113 Golgi cells for the hemisphere. (**B**) Computational model validation against the experimental data for the vermis and hemisphere. The plot shows the distribution of normalized response amplitude experiments (N_2a_ peak amplitude) and simulations using the same stimulation frequency. The time points considered for model validation are 1, 2, 10, 20, 25, and 35 s. The simulated frequencies are the same as those used in the experiments. The black lines are linear fittings (vermis: slope 0.81, R^2^ 0.73, *p* < 0.001; hemisphere: slope 0.80; R^2^ 0.70, *p* < 0.001), the dashed lines are 95% confidence intervals.

**Figure 6 cells-11-01047-f006:**
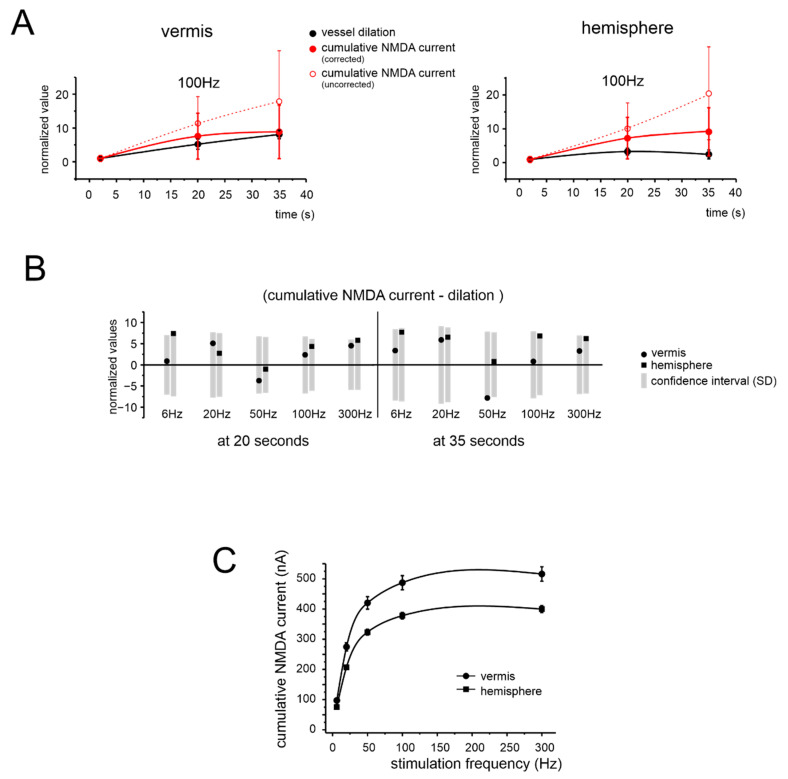
Comparison of simulated NMDA currents in granule cells and blood vessel changes. (**A**) Time course of the granule cells’ cumulative NMDA current during 100 Hz stimulation and of the corresponding vessel response, for the vermis and hemisphere. The plot shows both the corrected and uncorrected NMDA component (see text for details). All data are normalized to amplitude at 2 s. (**B**) The bar graph shows the difference between the normalized values of simulated NMDA components and vessel dilation at 20 and 35 s at all tested frequencies, for the vermis and hemisphere. The grey rectangles represent the normalized confidence interval (±1SD) yielded by model simulations of the NMDA current. Notice that all data points fall within the confidence interval, both for the vermis and hemisphere. (**C**) Frequency dependence of the cumulative NMDA currents measured at 35 s, for the vermis and hemisphere. This panel should be compared to the corresponding one for vasodilation (cf. Figure 2B).

**Figure 7 cells-11-01047-f007:**
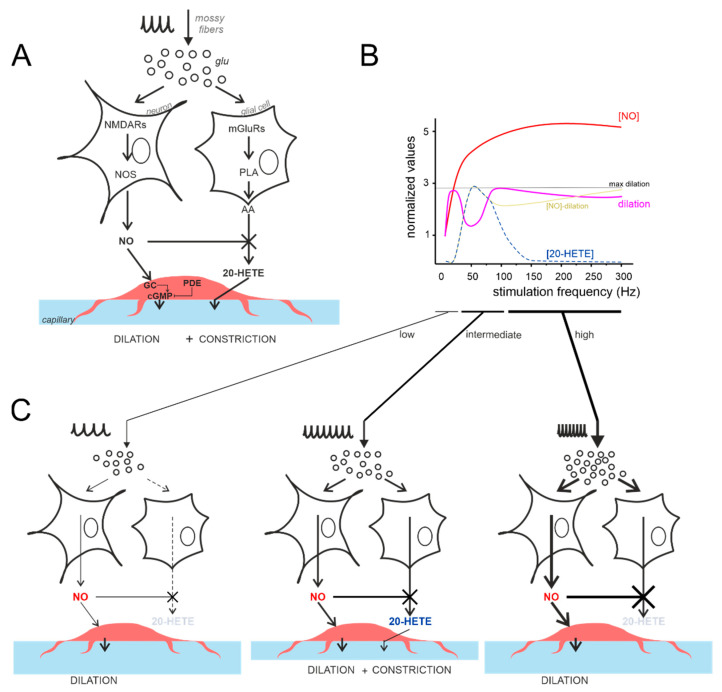
The vasodilation–vasoconstriction competition hypothesis. (**A**) Schematic illustration of the main players involved in NO and 20-HETE release, according to [21]. Vasodilation is mediated by the NMDAR-NOS-NO pathway, while vasoconstriction is mediated by the mGluR-PLA-20-HETE pathway. Notice that NO inhibits 20-HETE synthesis [8]. PLA, phospholipase A; GC, guanylyl cyclase; PDE, phosphodiesterase. (**B**) Schematic illustration of the competition between NO and 20-HETE in determining vessel diameter changes. [NO] (red) is directly proportional to the simulated NMDA current (not shown), while capillary dilation (violet) is provided by experimental measurements (the difference between the two curves is in yellow). The 20-HETE is specular to the NVC inflection at intermediate frequencies (more details are given in text). (**C**) The differential engagement on the NO and 20-HETE pathways are shown for low, intermediate, and high stimulus frequencies (note the different thickness of the arrows).

## Data Availability

All the materials related to the paper are available from the corresponding authors upon reasonable request.

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
