# Peer review of "Non-Linear Frequency Dependence of Neurovascular Coupling in the Cerebellar Cortex Implies Vasodilation–Vasoconstriction Competition"

_cells, 2022, doi:10.3390/cells11061047_

Round 1

Reviewer 1 Report

General comments:

In this article, the authors investigated the neurovascular coupling (NVC)/ neuronal activity (NA) relationship using a range of input frequencies in acute mouse cerebellar slices of vermis and hemisphere. 

Their results suggest that the force/BOLD nonlinearity recorded from the cerebellum during motor task execution could, at least in part, be due to local non-linear NVC properties.

They also integrate the neurogenic and metabolic hypotheses of NVC. Finally, they further open the perspective of a dynamic microvessel diameter regulation, which requires consideration to interpret BOLD signals in fMRI recordings.

This manuscript provides some new and vital information about the NVC in the cerebellar cortex and its local variations.

I have some comments to strengthen this manuscript.

Specific comments:

Major

1.  mossy fiber stimulation (page 4, lines 149 and 150) “the stimulating electrode placed on the mossy fibers”: The white matter in the cerebellar cortex contains not only mossy fibers but also climbing fibers and Purkinje cell axons. Therefore, the white matter stimulation may not result in pure mossy fiber stimulation. You’d better add some notes about this possibility in Materials and methods or Discussion.

2.  mossy fiber stimulation (page 3,n lines 144 and 145) “Mossy fibers were stimulated with 15V stimuli (of 200µs duration/pulse; corresponding to 50µA given the electrode resistance) at 6Hz, 20Hz, 50Hz, 100Hz, 300Hz for 35s.”

Previous studies that recorded mossy fiber activities in behaving animals (e.g., Tomatsu et al. 2016, J Neurophysiol doi: 10.1152/jn.00530.2015) demonstrated that mossy fiber activities are much less than 200 Hz in most cases. Therefore, stimulation at 300Hz may be well beyond a physiological (i.e., normal) range of mossy fiber activities. I suggest adding a paragraph in the Discussion to consider 300Hz data separately from lower frequency data.

Minor

  1. page 4, line 159: “Neuronal activity was recorded as local field potential (LFP)” -> Neuronal activity was recorded as local field potential (LFP) (Fig. 3)
  2. page 5, line 201: “29.415 gloms” -> Why is the number decimal? Should it be an integer?
  3. page 5, line 200 (384.000), line 201 (29.415), line 205 (630.419), line 206 (1.733), line 206 (48.291): Are all the digits relevant for the simulation? If so, is the model chaotic? In other words, does a slight difference in these parameters make a big difference in the results?
  4. page 5, lines 240 and 241: “Synaptic activation determines the release of vasoactive molecules.” If possible, it would be beneficial to give some examples of candidates of the vasoactive molecules.
  5. page 8, Figure 2: Use hatched-line for the legends of vermis data.
  6. page 9, Figure 3A You need calibration for the photographs.
  7. page 10, line 378: “N2b” lower case for 2b.
  8. page 11, line 405 “corrected by PDE”: Readers would appreciate more explanation for “the correction.”

Author Response

In this article, the authors investigated the neurovascular coupling (NVC)/ neuronal activity (NA) relationship using a range of input frequencies in acute mouse cerebellar slices of vermis and hemisphere. 

Their results suggest that the force/BOLD nonlinearity recorded from the cerebellum during motor task execution could, at least in part, be due to local non-linear NVC properties.

They also integrate the neurogenic and metabolic hypotheses of NVC. Finally, they further open the perspective of a dynamic microvessel diameter regulation, which requires consideration to interpret BOLD signals in fMRI recordings.

This manuscript provides some new and vital information about the NVC in the cerebellar cortex and its local variations.

I have some comments to strengthen this manuscript.

We thank the Reviewer for the positive evaluation of our work and the helpful comments. We provide a point-by-point reply below.

Specific comments:

Major

  1. mossy fiber stimulation (page 4, lines 149 and 150) “the stimulating electrode placed on the mossy fibers”: The white matter in the cerebellar cortex contains not only mossy fibers but also climbing fibers and Purkinje cell axons. Therefore, the white matter stimulation may not result in pure mossy fiber stimulation. You’d better add some notes about this possibility in Materials and methods or Discussion.

The Reviewer is right, the white matter bundle also contains climbing fibers and Purkinje cells axons. Nevertheless, in our case, their contribution is negligible for the following reasons. Climbing fibers make synaptic contacts with Purkinje cells, which do not contribute to the responses recorded in the granular layer and analyzed in this study. Purkinje cells axons contact deep cerebellar nuclei neurons, which again do not contribute to the granular layer response. Therefore, the only effect on granular layer responses to white matter bundle stimulation in our experiments is provided by mossy fibers activation. This is now reported in the Material and Methods section.

  1. mossy fiber stimulation (page 3,n lines 144 and 145) “Mossy fibers were stimulated with 15V stimuli (of 200µs duration/pulse; corresponding to 50µA given the electrode resistance) at 6Hz, 20Hz, 50Hz, 100Hz, 300Hz for 35s.”

Previous studies that recorded mossy fiber activities in behaving animals (e.g., Tomatsu et al. 2016, J Neurophysiol doi: 10.1152/jn.00530.2015) demonstrated that mossy fiber activities are much less than 200 Hz in most cases. Therefore, stimulation at 300Hz may be well beyond a physiological (i.e., normal) range of mossy fiber activities. I suggest adding a paragraph in the Discussion to consider 300Hz data separately from lower frequency data.

The Reviewer is right, the average frequency of mossy fiber discharge is below 200Hz. However, much higher frequencies have been described as instantaneous frequencies, with brief burst at more than 300Hz. For this reason, we also tested the ultra-fast band at 300Hz, though its physiological significance is less clear. In any case, the main finding of our paper concerns the inflection in capillary dilations at frequencies of 50-100Hz, which are physiologically relevant. These observations and relevant references are now reported in the Discussion and Supplementary Methods.

Rancz EA, Ishikawa T, Duguid I, et al. High-fidelity transmission of sensory information by single cerebellar mossy fibre boutons. Nature 2007; 450: 1245-1248. 2007/12/22. DOI: 10.1038/nature05995.

van Beugen BJ, Gao Z, Boele HJ, et al. High frequency burst firing of granule cells ensures transmission at the parallel fiber to purkinje cell synapse at the cost of temporal coding. Front Neural Circuits 2013; 7: 95. 2013/06/05. DOI: 10.3389/fncir.2013.00095.

Delvendahl I and Hallermann S. The Cerebellar Mossy Fiber Synapse as a Model for High-Frequency Transmission in the Mammalian CNS. Trends Neurosci 2016; 39: 722-737. 2016/10/25. DOI: 10.1016/j.tins.2016.09.006.

Minor

1. page 4, line 159: “Neuronal activity was recorded as local field potential (LFP)” -> Neuronal activity was recorded as local field potential (LFP) (Fig. 3)

Fig.3 is now cited in the sentence, now at line 166 (page 4).

2. page 5, line 201: “29.415 gloms” -> Why is the number decimal? Should it be an integer?

We thank the Reviewer for noticing this error. The use of the decimal separator (.) is of course wrong, the correct number writing is 29,415 or simply 29415 (as now reported in the text).

3. page 5, line 200 (384.000), line 201 (29.415), line 205 (630.419), line 206 (1.733), line 206 (48.291): Are all the digits relevant for the simulation? If so, is the model chaotic? In other words, does a slight difference in these parameters make a big difference in the results?

Similarly, the “point” here was incorrectly used to indicate thousands and not decimals. The text has been consequently amended.

4. page 5, lines 240 and 241: “Synaptic activation determines the release of vasoactive molecules.” If possible, it would be beneficial to give some examples of candidates of the vasoactive molecules.

As summarized in the Introduction, the vasoactive molecules reported to act as main mediators of vasodilation and vasoconstriction in the granular layer are nitric oxide and 20-HETE. This is now explicitly indicated at the beginning of the Results section, as suggested by the Reviewer.

5. page 8, Figure 2: Use hatched-line for the legends of vermis data.

The legend in the Figure panels B, C, E has been corrected using the hatched line.

6. page 9, Figure 3A You need calibration for the photographs.

We thank the Reviewer again for noticing this. We added the scale bars to the pictures in Fig.3A and integrated the corresponding legend.

7. page 10, line 378: “N2b” lower case for 2b.

We corrected with the lower case for 2b.

8. page 11, line 405 “corrected by PDE”: Readers would appreciate more explanation for “the correction.”

We agree with the Reviewer that the correction for PDE action needed further specifications. The text has been integrated as follows:

“Vasodilation in the granular layer is mediated by NMDAR-dependent production of NO (Mapelli et al., 2017) acting on guanylyl cyclase (GC) in pericytes and correlating almost linearly with cGMP levels (Batchelor et al., 2010). However, cGMP levels are known to decrease due to the action of phosphodiesterase (PDE) that activates with a slow time constant (around 20 seconds). This eventually decreases cGMP concentration by 26.4% and 59.9% after 20s and 35s, respectively (Batchelor et al., 2010). Estimating the impact of NMDAR activation on NVC then requires a correction for the PDE action”.

Reviewer 2 Report

The current manuscript by Gagliano et al investigated the non-linear frequency dependence of NVC in the cerebellar cortex using ex vivo model and computational model. The authors reported that capillary dilation varied depending on the input frequencies of activating mossy fiber. Furthermore, their computational model confirmed the role of NMDAR in eliciting capillary dilation, but additional mechanisms might be involved in explaining the frequency dependency of the responses.

These are interesting findings and this study opens more questions regarding the competing role of vasodilation and vasoconstriction in NVC and its mechanistic basis. Below are my comments.

  1. The authors mentioned about the dominant role of NO-pathway in eliciting vasodilation and the 20-HETE pathway in causing vasoconstriction as others have suggested in different brain regions, it would strengthen the study if the authors could provide some pharmacology where NO (or cGMP) and 20-HETE are manipulated at different frequencies.
  2. Why did the authors use 95% O2 instead of 20%?
  3. Giving that the diameter of these capillaries are ~2.75 (Vermis) and ~2.47 (Hemisphere) and the % change of the response is between 5-10%, the absolute value changes for the diameter is ~0.2-0.5um. How confident are the authors in accurately detecting these changes?
  4. Could you provide the onset and time-to-peak of the response?
  5. Could you elaborate a bit more about your criteria for calculating the slope (1-5s) and its significance?
  6. I think it would be a bit easier to see the differences if the data in Fig 2A is plotted as bar graphs.

Author Response

The current manuscript by Gagliano et al investigated the non-linear frequency dependence of NVC in the cerebellar cortex using ex vivo model and computational model. The authors reported that capillary dilation varied depending on the input frequencies of activating mossy fiber. Furthermore, their computational model confirmed the role of NMDAR in eliciting capillary dilation, but additional mechanisms might be involved in explaining the frequency dependency of the responses.

These are interesting findings and this study opens more questions regarding the competing role of vasodilation and vasoconstriction in NVC and its mechanistic basis. Below are my comments.

We thank the reviewer for the helpful comments. We provide a point-by-point reply below.

  1. The authors mentioned about the dominant role of NO-pathway in eliciting vasodilation and the 20-HETE pathway in causing vasoconstriction as others have suggested in different brain regions, it would strengthen the study if the authors could provide some pharmacology where NO (or cGMP) and 20-HETE are manipulated at different frequencies.

Concerning the NO-pathway, it has been reported that the NOS enzyme is calcium dependent (Salerno and Ghosh, 2009) and, in the cerebellum granular layer, requires the NMDAR activation (Mapelli et la., 2017). The increase in the stimulus frequency causes a build-up of NMDAR-mediated current (as evident also in the results shown in the manuscript), which brings about a calcium ions influx which increases the NOS activation and boosts NO production. Also in literature, it has already been reported that the increase in NO concentration leads to an increase in cGMP production (Batchelor et al., 2010). A change in the stimulation frequency can be easily considered as a change in NO concentration. Therefore, we can conclude that an increase in stimulation frequency is likely to determine a proportional increase in NO and, consequently, cGMP production.

Concerning 20-HETE, there are no reports in literature addressing the kinetics of its production, only the involved biochemical pathways. This is surely a topic that we hope will gain more attention in the future, considering that our results hint for a complex and dynamic interaction between vasoactive agents. Unfortunately, there are no available methods to measure 20-HETE production in slices during synaptic activation (e.g., fluorescent dyes to measure changes in 20-HETE concentration in the extracellular space).

2. Why did the authors use 95% O2 instead of 20%?

In vivo, neural tissue receives oxygenation from the blood circulation. In the slice condition, blood perfusion is absent in the vessel within the tissue, therefore the oxygenation is maintained at an optimal level in the extracellular solution in which the slices are kept. A good level of oxygenation in the blood is around (or even above) 95%. Indeed, while 20% is the percentage of O2 in the air we breathe, the blood can reach much higher saturation levels thanks to the efficiency of hemoglobin binding O2 molecules.

3. Giving that the diameter of these capillaries are ~2.75 (Vermis) and ~2.47 (Hemisphere) and the % change of the response is between 5-10%, the absolute value changes for the diameter is ~0.2-0.5um. How confident are the authors in accurately detecting these changes?

The measures were performed on images obtained using a 60X objective, allowing an adequate magnification to measure changes below the micron level. These measurements were obtained using the ImageJ software which allows to optimize the image contrast, allowing to identify even the inner walls of the vessel. Consider that in the example of the average vermis capillary diameter of 2.75 micron, the corresponding length of the measuring bar in ImageJ (as in Fig.1B) is of 30.5556 arbitrary units. A 5% increase in the diameter results in a shift to 32.0833 arbitrary units, a change that is clearly measurable with this tools. Moreover, this procedure is common and accepted practice when measuring microvessels diameter changes. A reference and a sentence have been added to the Methods.

4. Could you provide the onset and time-to-peak of the response?

The time-to-peak of vessels dilation is already reported as “time at which the maximum dilation is reached”, now at lines 299-300: “The time at which the maximum dilation was reached varied for each vessel, with an average of 23.7±1.5s in the vermis and 21.4±1.4s in the hemisphere (n=50 and 49, respectively, p=0.274)”.

Concerning the onset of dilation, the acquisition rate of 1 image per second is not sufficient to precisely measure the time at which the dilation starts. In the majority of the cases, at 1s after the stimulation onset we already measured a 0.3-0.4% of dilation (see Fig.1C). For this reason, we did not provide in the text a value for dilation onset. Since NO production is known to peak at about 200ms after MFs stimulation (see Discussion at line 553), it is likely that dilation onset is below the second and therefore not detectable in our case.

5. Could you elaborate a bit more about your criteria for calculating the slope (1-5s) and its significance?

The slope was simply calculated as the actual slope of the line passing through the points representing the capillaries dilations at 1s and 5s. The significance of this parameter is due to the time-range (few seconds) which is more comparable to BOLD signals measurements than 35s. Due to the slice condition, the maximum dilation was reached on average after 20s. It is likely that in vivo the time scale might be much shorter. The fact that we could trace back to the 5s scale a significant parameter correlating well with the maximum dilation makes our findings significant to the interpretation of the BOLD signal in fMRI scans. These concepts have been further integrated in the text in the Methods and Discussion.

6. I think it would be a bit easier to see the differences if the data in Fig 2A is plotted as bar graphs.

Figure 2A has been integrated with the bar graphs, as suggested by the Reviewer.

Round 2

Reviewer 2 Report

I think the manuscript is sufficient for publication.